# Protective Effects of *Sophorae tonkinensis* Gagnep. (Fabaceae) Radix et Rhizoma Water Extract on Carbon Tetrachloride-Induced Acute Liver Injury

**DOI:** 10.3390/molecules27248650

**Published:** 2022-12-07

**Authors:** Congcong Zhou, Aijing Liu, Gengsong Liu, Cheng Yang, Qiaoyan Zhou, Haizhu Li, Hongchun Yang, Mengmeng Yang, Gonghe Li, Hongbin Si, Changbo Ou

**Affiliations:** 1College of Animal Science and Technology, Guangxi University, Nanning 530004, China; 2College of Veterinary Medicine, Qingdao Agricultural University, Qingdao 266109, China; 3Guangxi Zhuang Autonomous Region Engineering Research Center of Veterinary Biologics, Nanning 530004, China; 4Guangxi Key Laboratory of Animal Reproduction, Breeding and Disease Control, Nanning 530004, China

**Keywords:** *Sophorae tonkinensis* Radix et Rhizoma, acute liver injury, traditional Chinese medicine, apoptosis, carbon tetrachloride, inflammation

## Abstract

*Sophorae tonkinensis* Radix et Rhizoma (STR) is a traditional Chinese herbal medicine. STR can reduce aminotransferase activity; however, the specific mechanism remains unclear. Here, we explored the potential therapeutic effects and hepatoprotective mechanism of STR on liver damage in mice. The chemical characteristics of the extract were characterized using ultra-high-performance liquid chromatography-tandem mass spectrometry fingerprinting, and its antioxidant capacity was verified using free radical scavenging tests. Forty-eight Kunming mice were randomly assigned into six groups. The model was made after the corresponding drug was given. The results showed that the STR water extract pretreatment significantly reduced serum aminotransferase and related liver function indicators compared with that in the model group. Furthermore, the STR water extract pretreatment significantly inhibited the apoptosis of liver cells, the level of liver high-mobility group box 1 (HMGB1), and inflammatory factors in hepatic tissue compared with that in the model group, and significantly downregulated the levels of toll-like receptor 4 (TLR4), Myeloid differentiation factor 88 (MyD88), and nuclear factor kappa B (NF-κB) compared with those in the model group. Overall, the STR water extract exerted a significant protective effect on CCL_4_-induced acute liver injury in this study, and the accurate active ingredients of the STR water extract will be explored in the near future.

## 1. Introduction

In recent years, liver-related diseases have emerged as global diseases with high morbidity and mortality [1,2]. Acute liver injury is a main cause of liver disease. The liver is one of the largest and most important organs in the body [3], which participates in toxicant transformation and plays an essential role in maintaining the basic metabolism of the body and other functions [4,5]. When the liver is attacked by various pathogenic factors, such as infectious and noninfectious factors, hepatocytes become damaged or necrotic; thus, abnormal liver function leads to acute liver injury [6,7]. If liver injury is not immediately treated, it may develop into acute liver failure, cirrhosis, and other serious diseases, eventually endangering life. Several factors, including viruses, drugs, high-fat diets, immune responses, and liver ischemia and reperfusion, are common factors in acute liver injury [8]. Acute liver injury is accompanied by oxidative stress, inflammation, apoptosis, and autophagy, which likely involves the release of high-mobility group box 1 (HMGB1) [9,10,11]. HMGB1 plays an important role in many liver diseases, such as alcoholic liver disease, drug-induced liver injury, and liver ischemia-reperfusion [12,13,14,15]. HMGB1 regulates the nuclear factor kappa B (NF-κB) and mitogen-activated protein kinase (MAPK) pathways through toll-like receptors (TLRs) and modulates the expression of inflammatory factors [16].

Carbon tetrachloride (CCL_4_)-induced acute liver injury is a widely used model in the study of liver damage because it replicates acute chemical liver injury in humans [17,18,19]. In the liver, CCL_4_ is metabolized to trichloromethyl radicals (CCL_3_*) by monooxygenases of the cytochrome P450 family. CCL_3_* combines with oxygen to form trichloromethyl peroxy radicals (CCl_3_OO*), which then react with biological macromolecules, such as nucleic acids, proteins, and lipids, thereby damaging key cellular processes, resulting in altered lipid metabolism, decreased membrane permeability, the degeneration and necrosis of liver cells, and eventually systemic liver injury characterized by inflammation, fibrosis, cirrhosis, and hepatocellular carcinoma [8]. In acute liver injury in mice, HMGB1 plays an important role, and when the expression of HMGB1 is inhibited, the oxidative stress and inflammatory response of the liver are also inhibited [20]. Similarly, a recent study has suggested that chloroquine pretreatment inhibits HMGB1-mediated inflammatory responses to improve CCL_4_-induced acute liver injury [21].

Many herbal extracts prevent and protect against liver damage, and have been widely used in clinical practice. Chinese herbal medicine is generally considered to be less toxic and safer than Western medicine; therefore, it is used in various fields. *Sophorae tonkinensis* Radix et Rhizoma (STR), known as “Shandougen” in Chinese, is the dried root of *Sophora tonkinensis* Gagnep, a legume plant that is mainly distributed in Guangxi, Guangdong, Sichuan, Hunan, and other regions in China, and is occasionally distributed in Japan. STR is a traditional Chinese herbal medicine widely used in folk medicine and mainly contains alkaloids [22,23,24], flavonoids [25], and other active ingredients [26,27]. In traditional Chinese medicine (TCM), STR represents bitterness, coldness, and a return to the lung and stomach meridian, and exerts heat clearing and detoxifying effects, reduces swelling, and supports the pharynx. Furthermore, STR is often used in the treatment of fire and poison stasis, throat swelling and pain, mouth and tongue sores, and other symptoms [28]. The polysaccharides from STR exert a significant antioxidant effect, which may significantly reduce the oxidative damage to the liver caused by acetaminophen [29]. STR extracts may exert therapeutic effects on non-alcoholic fatty liver disease in mice [30]. Gan et al. showed that oxymatrine, the main active ingredient of STR, inhibits activation of the HMGB1/TLR4 pathway and nuclear translocation of NF-κB to reduce neuroinflammation in Parkinson’s disease compared with that in controls [31].

To investigate the hepatoprotective effect and mechanism of STR, we established a mouse model of acute liver injury induced by CCL_4_. The hepatoprotective effect of STR was further evaluated using biochemical index detection, liver histology examination, and related gene expression analysis, and the mechanism of its possible hepatoprotective effects was explored.

## 2. Materials and Methods

### 2.1. Materials and Reagents

STR was purchased from a TCM store (Nanning, China). CCL_4_ (purity ≥ 99%) was purchased from Hengxing Reagent (Tianjin, China). L-ascorbic acid (purity > 99%) and potassium persulfate (purity > 99.5%) were purchased from Shanghai Macklin Biochemical Technology Co., Ltd. (Shanghai, China) and 1,1-diphenyl-2-picryhydrazyl free radical (DPPH) (Lot No. 1898-66-4) was purchased from Tixiai Chemical Industry Development Co., Ltd. (Shanghai, China). 2,2′-azino-bis(3-ethylbenzothiazoline-6-sulfonate) (ABTS; Lot No. 30931-67-0) was purchased from Beijing Biotopped Technology Co., Ltd. (Beijing, China). The aspartate aminotransferase (AST) (Lot No. C010-2-1), alanine aminotransferase (ALT) (Lot No. C009-2-1), and protein quantification (Lot No. A045-4) kits were purchased from Nanjing Jiancheng Biotechnology Co., Ltd. (Nanjing, China). The total bilirubin (TBIL) content detection (Lot No. BYSH-1226W), alkaline phosphatase (ALP) activity assay (Lot No. BYSH-0904F), HMGB1 enzyme-linked immunosorbent assay (ELISA) (Lot No. R20318), interleukin (IL)-1β ELISA (Lot No. R20174), IL-6 ELISA (Lot No. R20188), and tumor necrosis factor-α (TNF-α) ELISA (Lot No. R20852) kits were purchased from Nanjing Boyan Biotechnology Co., Ltd. (Nanjing, China). TRIzol reagent was purchased from Thermo Fisher Scientific (Waltham, MA, USA), and the HiScript II One-Step qRT-PCR SYBR Green kit was purchased from Vazyme Biotechnology Co., Ltd. (Nanjing, China). All other reagents used in the experiments were of analytical purity.

### 2.2. Experimental Animals

Forty-eight five-week-old healthy Kunming mice (male) weighing 34 ± 36 g were purchased from Guangxi Medical University (Guangxi, China). All experimental mice were randomly divided into six groups, with eight mice in each group, and the experiments were performed seven days after adaptive rearing. The room temperature was 22 ± 2 °C, the air humidity was 55 ± 20%, and the light was 12 h/day. The mice were allowed to drink freely, and each mouse ate 3.5 g food per day. The experiments were performed in strict accordance with the standards of the Animal Care and Welfare Committee of Guangxi University (Approval No. GXU-2022-082).

### 2.3. Preparation of the Test Drug

Dried STR (100 g) (Figure 1) was washed with deionized water, and STR was then extracted three times using 10×, 8×, and 8× deionized water (weight/volume) for 1 h each time. The combined extraction and supernatant were collected after centrifugation at 3500 rpm for 10 min. The supernatant was then concentrated and dried in a vacuum freeze-dryer (Shanghai Lichen Bangxi Instrument Technology Co., Ltd., Shanghai, China). The mass of the final lyophilized powder of STR (100 g) was 6.67 g and the extraction rate was 6.67%. Physiological saline was used to prepare concentrations of 80 mg/mL, 40 mg/mL, and 20 mg/mL labeled as a high-, medium-, and low-dosage extracts of STR, respectively. The STR solutions were placed in a 4 °C refrigerator until further use. Bifendate (Lot No. H33021305) was purchased from Wanbond Pharmaceutical Group Co., Ltd. (Wenling, China), diluted with normal saline, and prepared as an aqueous solution of 15 mg/mL for experiments.

### 2.4. Ultra-High-Performance Liquid Chromatography-Tandem Mass Spectrometry (UPLC-MS/MS) Analysis of STR

To evaluate the material basis of the pharmacological effects of STR, the STR aqueous extract was diluted 100 times with absolute ethanol and centrifuged at 12,000× *g* at 4 °C for 10 min to obtain the supernatant. The supernatant was then filtered through a 0.22 µm organic nylon filter membrane and the filtrate was collected. The UPLC separations were performed using an ACQUITY UPLC BEH C18 column (2.1 × 5 mm i.l, 1.7 µm; Waters Inc., Milford, MA, USA). The mobile phase was 0.1% formic acid water and methanol, and the elution procedure was as follows: 0–13 min, 5% B–100% B, 100% B for 3 min, 16–19 min, and 100% B–5% B. The column temperature was maintained at 30 °C and eluted at a flow rate of 0.3 mL/min. The UPLC chromatograms were recorded and MS spectra with an *m*/*z* scan range of 70–1000 were obtained in both the positive and negative modes.

### 2.5. Analysis of STR Antioxidant Activity In Vitro

#### 2.5.1. DPPH Radical Scavenging Test

The DPPH radical scavenging test was performed as previously described [32], with minor adjustments. Briefly, 50 µL of the sample solution at different concentrations was added to a microplate with 50 µL of DPPH stock solution, and absorbance (A2) at 517 nm was measured after 30 min of light avoidance reaction at room temperature. The DPPH radical-scavenging rate of the extract (E) was calculated as follows:E (%) = [1 − (A2 − A1)/A0] × 100%
where A2 represents the absorbance of the sample, A1 represents the absorbance of the sample without the DPPH working solution, and A0 represents the absorbance of the blank wells.

#### 2.5.2. ABTS Radical Scavenging Test

The ABTS radical scavenging test was performed as previously described [33], with minor adjustments. Briefly, 50 µL of the sample solution at different concentrations was added to a microplate with 150 µL of ABTS working solution. The absorbance (Ai) at 734 nm was measured after 6 min of light avoidance reaction at room temperature. The ABTS radical-scavenging rate of the extract was calculated using the following formula:E (%) = [1 − (Ai − Aj)/A0] × 100%
where A0 represents the absorbance of the blank well, Ai represents the absorbance of the sample, and Aj represents the absorbance of the sample without ABTS (the ABTS working solution was replaced with absolute ethanol in the reaction wells).

#### 2.5.3. Hydroxyl Radical Scavenging Test

The hydroxyl radical scavenging test was performed as previously described [32], with minor modifications. Firstly, 50 µL of each concentration of the sample, negative control (water), and positive control (vitamin C) were added to a 96-well microplate. Next, 50 µL of ferrous sulfate solution, 50 µL of salicylic acid-ethanol solution, and 50 µL of H_2_O_2_ solution were added to all wells. The reaction occurred at 37 °C for 30 min after mixing, and the absorbance value of the sample at 510 nm was determined. The hydroxyl radical-scavenging activity of the extract was calculated as follows:E (%) = [A0 − (A − Ax)]/A0 × 100%
where A0 represents the absorbance of the blank well, A represents the absorbance of the sample solution, and Ax represents the absorbance of the sample or its derivative itself (without H_2_O_2_).

### 2.6. Animal Treatment

Forty-eight mice were randomly divided into six groups (*n* = 8 each): normal; model; Western medicine control; and high-, medium-, and low-dose groups of STR (STR-H, STR-M, STR-L, respectively). The normal and model groups were administered saline by gavage, whereas the positive control group was administered 150 mg/kg bifendate aqueous solution and 10 mL/kg body weight (BW) by gavage. According to the “dose translation from animal to human studies revised” [34], the human equivalent dose (HED) (mg/kg) = animal dose (mg/kg) × animal Km/human Km. Therefore, the mouse dose of STR (g/kg) = HED × human Km (37)/mouse Km (3). As the Chinese Pharmacopoeia requires a dose of 3–6 g/60 kg BW, the mouse dose of STR (g/kg) = 6 g/60 kg × (37/3) = 1.23 g/kg and the mouse dose of STR extract (mg/kg) = 1.23 g × 6.67% = 82 mg/kg BW (6.67% refers to the yield of water extract of STR). In this study, the three groups of mice were administered STR via gavage at doses of 80 mg/kg, 40 mg/kg, and 20 mg/kg (10 mL/kg) once a day, and all groups were continuously gavaged for 14 days. After intragastric administration for 2 h on the last day, all groups, except the normal group, were injected with 0.2% CCL_4_ olive oil solution at 10 mL/kg BW to establish the liver injury model, whereas the normal group was injected with an olive oil solution without CCL_4_ as a control. The mice were then fasted but provided with a water supply for 16 h and then weighed. After anesthesia, blood was collected from the mice to prepare the serum. The liver was dissected after cervical dislocation, rinsed with ice-cold saline, and weighed. Half of the liver tissue was routinely fixed and embedded in wax blocks and the other half was placed in liquid nitrogen for 1 min and then stored at −80 °C.

### 2.7. Histopathological Liver Examination

Simultaneously as the mice were sacrificed, a portion of the hepatic tissue was quickly collected and fixed in 4% paraformaldehyde. Next, a small piece of tissue was excised, embedded, dehydrated, made into paraffin blocks, sectioned (3–5 μm thick), and stained using hematoxylin and eosin (HE) for histopathological observation using a light microscope.

### 2.8. Serum Biochemistry

Peripheral blood was collected in a sterile tube by removing the eyeballs before the mice were sacrificed. After storing for 2–3 h at 4 °C, the serum was collected via centrifugation at 3000× *g* at 4 °C for 10 min and then stored in a refrigerator at −80 °C. Different biochemical kits were used to detect the serum AST, ALT, TBIL, and ALP levels according to the manufacturer’s instructions.

### 2.9. ELISA

Chopped liver tissue (100 mg) was mixed with 900 µL of phosphate-buffered saline (PBS) (containing protease inhibitors) and homogenized thoroughly on ice. To lyse the tissue further, the homogenate was broken using a low-temperature ultrasound. Finally, the homogenate was centrifuged at 5000× *g* for 5–10 min, and the supernatant was collected for cytokine measurement. The levels of inflammatory factors, such as HMGB1, IL-1β, IL-6, and TNF-α, in the liver tissues were detected using an ELISA kit according to the manufacturer’s instructions.

### 2.10. Real-Time Polymerase Chain Reaction (qRT-PCR)

Total RNA was extracted from the liver tissue using TRIzol reagent (Invitrogen, Waltham, MA, USA) according to the manufacturer’s instructions. PCR was performed using the HiScript II One Step qRT-PCR SYBR Green Kit (Vazyme) on a LightCycler 96 system (Roche, Basel, Switzerland) according to the manufacturer’s instructions. Glyceraldehyde 3-phosphate dehydrogenase was used as the reference gene to normalize gene expression. The fold-change in gene expression was calculated using the threshold cycle method (2 − ΔΔCT). The primer sequences (Sangon Biotech Co., Ltd., Shanghai, China) are listed in Table 1.

### 2.11. Hepatocyte Apoptosis

Real-time quantitative reverse transcription PCR (qRT-PCR) was used to detect the expression of apoptosis-related genes in hepatocytes.

### 2.12. Statistical Analysis

Data are expressed as mean ± standard deviation. Statistical analyses were conducted using SPSS Statistics 26 (IBM, Armonk, NY, USA), and one-way analysis of variance (ANOVA) was used to compare differences between groups, followed by Dunnett’s multiple comparison test. Statistical significance was set at *p* < 0.05.

## 3. Results

### 3.1. Quality Control Analysis of STR by UPLC-MS/MS

As shown in Figure 2, UPLC-MS/MS was used to predict the chemical composition of the STR water extract, and the spectra fingerprint of the STR water extract was established as a qualitative reference index.

### 3.2. Antioxidant Assay of STR In Vitro

Free radical scavenging tests were performed to evaluate the antioxidant capacity of STR water extract. As shown in Figure 3A,B, similar to the typical antioxidant vitamin C, STR exerted free radical scavenging effects for DPPH and ABTS. For the DPPH free radical, vitamin C maximally scavenged 100% DPPH at 1 mg/mL, whereas STR maximally scavenged 89.32% DPPH at the same concentration. For the ABTS free radical, vitamin C scavenged 99.65% ABTS at 0.25 mg/mL, whereas STR scavenged 99.88% at the same concentration. The ability of STR to scavenge hydroxyl radicals was relatively poor, scavenging only 35.12% at 4 mg/mL. However, these results indicate that STR presents a high antioxidant capacity.

### 3.3. Effect of STR on Liver Histopathology in Mice

To explore the effects of STR on liver histopathology in mice, we performed HE staining. The hepatic lobule structure was clear, the hepatic cords were arranged neatly, the morphology of the hepatocytes was normal, and the nuclei were obvious in the blank group (Figure 4A). In the model group, hepatocytes were severely damaged, with a large amount of fibrosis around the central vein, destruction of the structure of the hepatic lobules, disappearance of hepatic sinusoids, and pyknosis (Figure 4B). The structure of the hepatic cord in the biphenyl diester group was relatively clear, the hepatocyte structure was relatively complete, and a small amount of lymphocyte infiltration was observed in the hepatic sinusoids compared with that in the model group (Figure 4C). The high-dose group showed central venous congestion and exudative nodules (Figure 4D). The morphology in the medium-dose group was relatively improved compared with that in the model group, with a clear structure of the hepatic cord, but partial nuclear pyknosis and a small amount of inflammatory cell infiltration were observed (Figure 4E). In the low-dose group, the structure of the hepatic cord in the central vein was clear, the hepatic sinusoids were obvious, and the hepatocytes were relatively intact; however, a small number of hepatocyte nuclei were pyknotic (Figure 4F). In conclusion, the high-, medium-, and low-dose groups of STR demonstrated a protective effect against CCL_4_-induced liver injury.

### 3.4. Effect of STR on Liver Function Indicators

AST and ALT levels, which are important indicators of liver function, were used to evaluate liver damage. The model group showed significantly higher AST, ALT, ALP, and TBIL levels (*p* < 0.01) than the normal group (Figure 5). The high-dose STR (80 mg/kg), medium-dose STR (40 mg/kg), and positive drug STR groups showed significantly reduced AST and ALT levels (*p* < 0.01 or *p* < 0.05), whereas the medium dose-STR (40 mg/kg), low-dose STR (20 mg/kg), and positive drug group demonstrated significantly reduced TBIL and ALP levels (*p* < 0.01) compared with those in the model group. No significant differences were observed between the three dose groups of STR. These findings suggest that STR exerts a protective effect against CCL_4_-induced liver damage and significantly improves liver function.

### 3.5. STR Treatment Reduces the CCL_4_-Induced Inflammatory Response

HMGB1 is an inflammatory cytokine and key endogenous danger signal molecule in the nucleus [35]. TNF-α, IL-6, and IL-1β are common pro-inflammatory factors. Compared with that in mice from the normal control group, the hepatic tissue expression of HMGB1, TNF-α, IL-6, and IL-1β was significantly increased (*p* < 0.01) in the model group (Figure 6). Furthermore, compared with that in the model group, the medium-dose STR group (40 mg/kg) showed significantly reduced HMGB1 levels (*p* < 0.01). The liver TNF-α and IL-6 inflammatory factor levels in mice from the STR group treated with 40 and 20 mg/kg STR and the positive control group were significantly inhibited (*p* < 0.01 or *p* < 0.05), whereas the low-dose group (20 mg/kg) of STR was able to reduce the level of IL-1β (*p* < 0.05) compared with those in the model group. In summary, STR extract inhibited CCL_4_-induced liver inflammation responses, improved anti-inflammatory ability, and protected against CCL_4_-induced liver damage.

### 3.6. Effect of STR on CCL_4_-Induced Hepatocyte Apoptosis

As shown in Figure 7, compared with those in the normal group, liver p53, Bax, and caspase-3 expression levels were significantly increased after CCL_4_ treatment (*p* < 0.01 or *p* < 0.05). All STR treatment groups and the positive control group (bifendate) exhibited significantly reduced levels of apoptosis-related genes (*p* < 0.01 or *p* < 0.05) compared with those in the model group, indicating that STR extract may inhibit CCL_4_-induced apoptosis in hepatocytes.

### 3.7. Effects of STR on the TLR4/MyD88/NF-κB Pathway in CCL4-Induced Liver Injury

The pathogenesis of acute liver injury is closely related to the inflammatory response, and NF-κB is a classic inflammatory response pathway. The TLR4/MyD88/NF-κB cascade plays an indispensable role in acute liver injury [36,37]. Compared with that in the normal control group, TLR4, MyD88, and NF-κB expression levels were significantly increased after CCL_4_ treatment (*p* < 0.01; Figure 8), whereas the inhibitor of NF-κB alpha (IκBα) expression was reduced. Compared with that in the model group, all STR treatment groups and the positive control group (biphenyl diester) displayed reduced expression of TLR4, MyD88, and NF-κB, particularly in the low-dose group of STR (*p* < 0.01). The high and low doses of STR significantly increased the level of IκBα, thereby inhibiting the activation of NF-κB and the inflammatory response to CCL_4_-induced acute liver injury in mice.

## 4. Discussion

This study aimed to assess the protective utility of the STR water extract against CCl_4_-induced liver damage in mice. STR was extracted using the common, reliable, sensitive, and accurate technique. It was chosen for its therapeutic characteristics, which have been widely utilized as traditional medicine in local communities to treat a series of illnesses, including fire and poison stasis, throat swelling and pain, mouth and tongue sores, and other symptoms [28].

The STR water extract showed strong free radical scavenging ability in this study. It contained flavonoids [25], alkaloids [22,23,24], and polysaccharides and other compounds [26,27]. Botanical drugs have a variety of biological activities, such as immunity boosting, antibacterial, antioxidant, and anti-inflammatory activities. The literature has documented that part of the compounds have potential health-promoting properties [29].

The liver is the largest parenchymal organ in the human body, and liver-related diseases cause severe damage worldwide, killing more than one million people every year [1,2,38]. This demonstrates the urgency of developing treatments for liver disease. In this study, we provide data that support a protective effect of the aqueous extract of STR in CCL_4_-induced liver injury, which may effectively alleviate CCL_4_-induced acute liver injury in mice. Liver disease can be modeled using drugs, alcohol, aflatoxin, chemicals, immunosuppression, and other methods [8,39,40]. Herein, we used CCL_4_ to generate a liver disease model, and the symptoms, liver function, and pathological changes of this model were similar to those of human liver injury [8,17,19]. The CCl_3_OO* free radical produced by CCL_4_ metabolism and oxidation subsequently reacts with nucleic acids, proteins, and lipids in the body, thereby damaging liver cell membranes, disrupting liver function, and eventually leading to changes in liver lipid metabolism, degeneration, and the necrosis of hepatocytes [41]. In this study, we revealed that the aqueous extract of STR plays a hepatoprotective role by exerting anti-inflammatory effects and inhibiting apoptosis, indicating that STR is a potential therapeutic agent for liver injury.

When hepatocytes are damaged, the permeability of the liver cell membrane increases, and AST and ALT enter the blood, resulting in a significant increase in the serum levels of these enzymes, accompanied by an increase in TBIL and ALP. ALT is mainly present in the cytoplasm of hepatocytes and is a sensitive indicator of liver cell damage. In contrast, AST is mainly present in the mitochondria of hepatocytes, myocardium, and skeletal muscle. TBIL reflects the transformative metabolic function of the liver, whereas ALP is mainly distributed in the liver and is often elevated when hepatitis develops. These four indicators are commonly used to determine whether a drug exerts a liver-protective effect. Therefore, the serum biochemical indicators ALT, AST, TBIL, and ALP are important in evaluating liver functional damage [42]. In the present study, we found that the serum levels of ALT, AST, TBIL, and ALP in the model group were significantly higher than those in the normal group, and similarly, the pathological sections of liver tissue showed a larger amount of fibrosis around the central vein and local inflammatory cell infiltration, suggesting that the acute liver injury model was successfully established. Treatment with an aqueous extract of STR significantly improved liver injury caused by CCL_4_, suggesting that STR exerts a significant protective effect on CCL_4_-induced acute liver injury.

The pathogenesis of acute liver injury is closely related to the inflammatory response [43]. HMGB1 is an abundant and widely expressed DNA-binding protein involved in a variety of pathological and physiological processes [12,13,14]. When cells or tissues are damaged, HMGB1 is released from necrotic cells into the extracellular space or through activated immune cells. HMGB1 then acts as an inflammatory factor or signal through its specific ligands, TLR4, RAGE, and other pathways to activate downstream NF-κB and MAPK pathways. This cascade regulates the release of inflammatory mediators, such as TNF-α, IL-1β, and IL-6, and further aggravates the inflammatory response [16]. Inflammatory cytokines, including IL-1β, IL-6, and TNF-α, play an important role in the inflammatory process of acute tissue injury. When hepatic macrophages are damaged, TNF-α inflammatory factors are released first. Il-1β is the strongest inflammatory factor in the body and an important mediator of immune and inflammatory responses by promoting the expression of inflammatory cytokines and inducing cellular immune responses. IL-6 is a multifunctional cytokine that exerts pro- and anti-inflammatory effects depending on the immune response. IL-6 participates in the immune response by activating and regulating immune cells. However, the overexpression of IL-6 induces a severe inflammatory response [44]. The injection of HMGB1-neutralizing antibodies effectively inhibits CCL_4_- and acetaminophen-induced liver injury by inhibiting inflammation [12,20]. In the present study, pretreatment with STR significantly reduced the increase in HMGB1 caused by CCL_4_, inhibited inflammatory mediators such as TNF-α and IL-1β, and inhibited the inflammatory response induced by CCL_4_. Therefore, HMGB1 is a promising therapeutic target in the treatment of liver injury.

CCL_4_ is metabolized by the liver and releases a large amount of reactive oxygen species, leading to apoptosis, autophagy, inflammation, and other pathological reactions [20,45,46]. The mitochondrial pathway plays an important role in CCL_4_-induced apoptosis [47] and is regulated by both anti- and pro-apoptotic proteins [48]. Cytochrome C is released in response to mitochondrial stress and activates caspase-3 and caspase-9, thus leading to apoptosis [48]. The apoptosis-activating gene p53 activates the pro-apoptotic protein Bax, which may increase the permeability of the mitochondria and further lead to mitochondrial cytochrome C release and caspase activation, retriggering cell apoptosis [49]. Herein, the expression of Bax, p53, and caspase-3 was increased in the CCL_4_-treated group compared with that in the normal control group. The pro-apoptotic effects of CCL_4_ were inhibited by pretreatment with STR, indicating that STR blocked CCL_4_-induced apoptosis.

TLR4 is closely related to liver inflammation [1] and is an important recognition receptor on the cell surface. MyD88 is a key downstream signal of TLR4. The damage factors generated by external stimuli, such as HMGB1, can induce TLR4 activation and activate downstream MyD88 signaling, leading to the rapid activation of NF-κB and production of pro-inflammatory factors. TLR4/MyD88/NF-κB is an important signaling pathway involved in liver inflammation [50,51,52], and downregulation of the TLR4/MyD88/NF-κB pathway may significantly improve liver injury triggered by the inflammatory response, which is consistent with our results.

Lastly, due to the complexity of TCM components, limitations in this study remain. However, as shown in Figure 9, the aqueous extract of STR is a promising treatment to significantly improve CCL_4_-induced liver injury. Further studies are needed to explore the mechanism underlying the hepatoprotective effect of STR.

## 5. Conclusions

CCL_4_ is commonly used in hepatotoxic models, and we used CCL_4_ to establish toxic liver injury in this study. UPLC-MS/MS was used to identify the chemical composition of the STR water extract. In the present study, the STR extract showed significant free radical scavenging activity. Additionally, the decrease of liver function indexes and hepatocyte apoptosis level suggested that the STR extract had a significant hepatoprotective effect on CCL_4_-induced liver injury, which was further verified by histopathological changes and inflammatory factors. We concluded that the STR extract could be preliminarily used as a therapeutic drug in the treatment of toxic liver injury, and it is very necessary to further study its mechanism and pharmacokinetics.

## Figures and Tables

**Figure 1 molecules-27-08650-f001:**
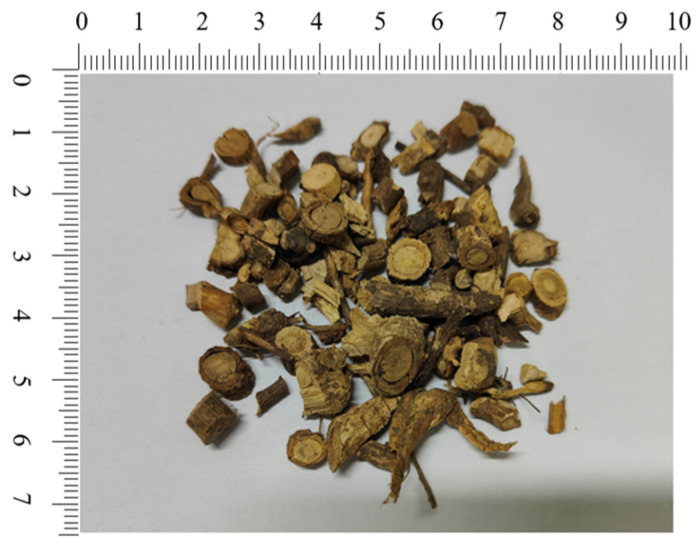
*Sophorae tonkinensis* Radix et Rhizoma (STR) used in the study.

**Figure 2 molecules-27-08650-f002:**
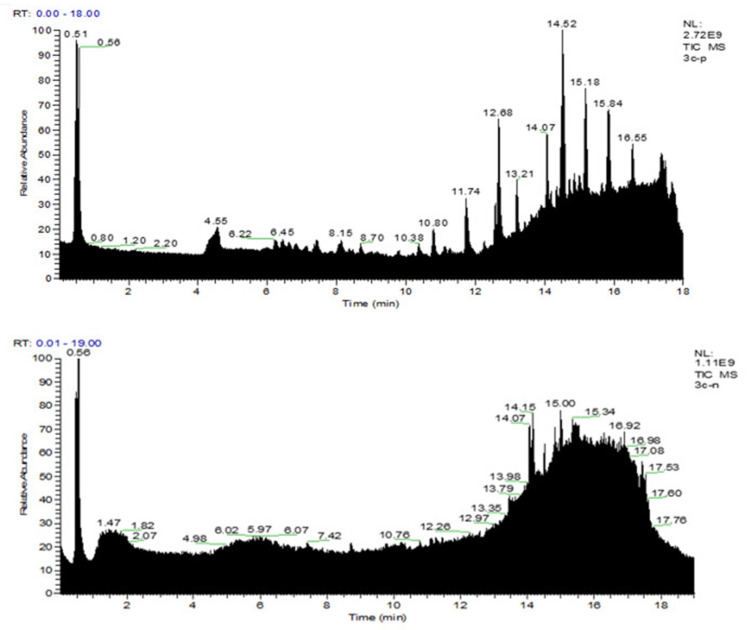
Ultra-high-performance liquid chromatography-tandem mass spectrometry (UPLC-MS/MS) analysis of a water extract of Sophorae tonkinensis Radix et Rhizoma (STR). UPLC-MS/MS spectra fingerprint of STR. Upper: positive mode (+); lower: negative mode (−).

**Figure 3 molecules-27-08650-f003:**
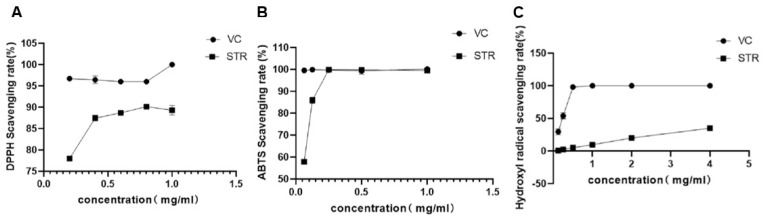
Antioxidant activity of STR in vitro. (**A**) 1,1-diphenyl-2-picryhydrazyl (DPPH) radical scavenging activity. (**B**) 2,2′-azino-bis (3-ethylbenzothiazoline-6-sulfonate) (ABTS) radical scavenging activity. (**C**) Hydroxyl radical scavenging activity. Data are expressed as mean ± standard deviation (SD), *n* = 3.

**Figure 4 molecules-27-08650-f004:**
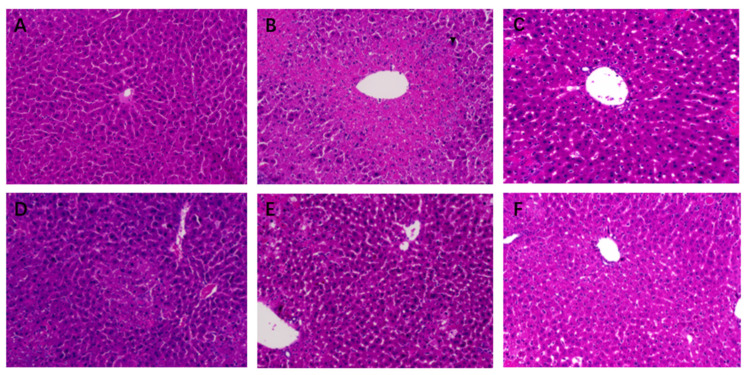
Effects of STR on histopathological changes in acute liver injury (ALI) mice. (**A**) Normal (control); (**B**) model; (**C**) bifendate control (150 mg/kg; positive control); (**D**) STR-high (80 mg/kg); (**E**) STR-medium (40 mg/kg); and (**F**) STR-low (20 mg/kg) groups. Magnification: 200×.

**Figure 5 molecules-27-08650-f005:**
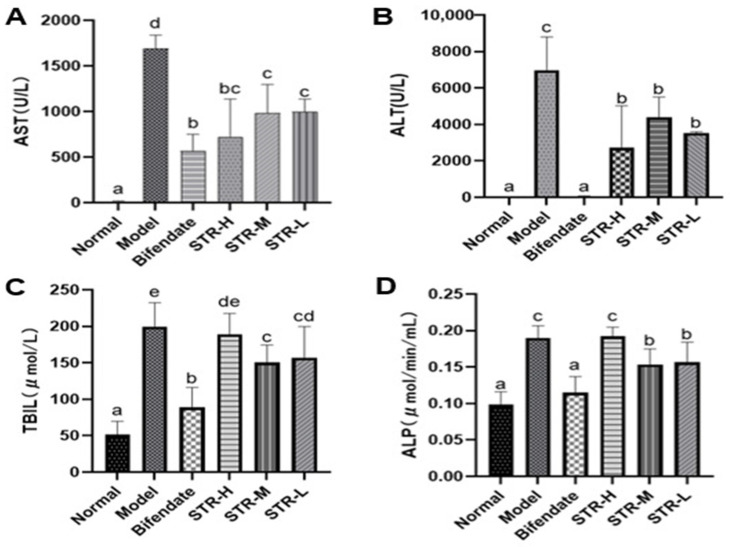
STR reduces levels of liver damage markers. (**A**) AST; (**B**) ALT; (**C**) TBIL; and (**D**) ALP levels as measured in mice serum by biochemical methods. Data are expressed as mean ± SD, *n* = 8. Different letters indicate statistically significant differences between groups at (*p* < 0.05), one-way ANOVA.

**Figure 6 molecules-27-08650-f006:**
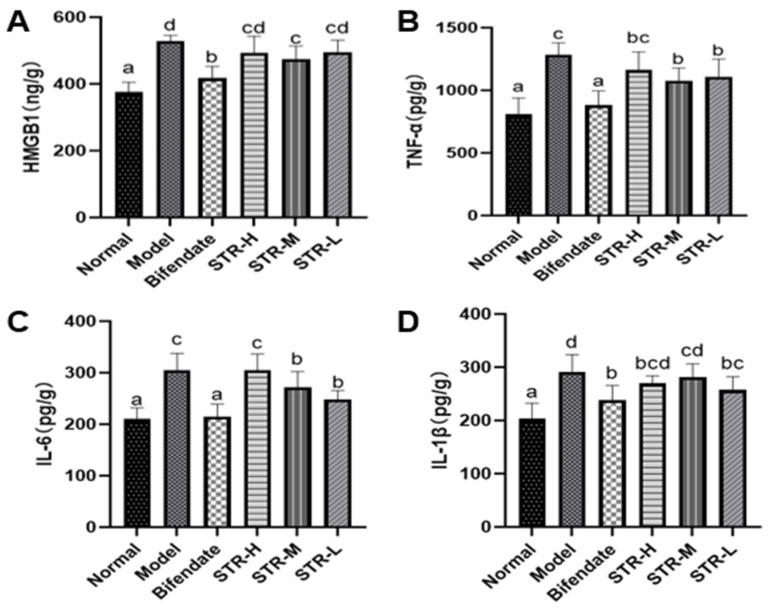
STR reduces the level of liver HMGB1, TNF-α, IL-6, and IL-1β in the mouse model of ALI induced by CCL_4_. (**A**) HMGB1; (**B**) TNF-α; (**C**) IL-6; and (**D**) IL-1β levels measured using enzyme-linked immunosorbent assay. Data are expressed as mean ± SD, *n* = 8. Different letters indicate statistically significant differences between groups at (*p* < 0.05), one-way ANOVA.

**Figure 7 molecules-27-08650-f007:**
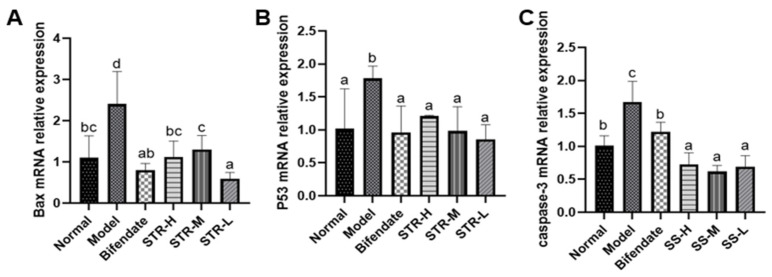
STR inhibits hepatocyte apoptosis induced by CCL_4_. (**A**) Bax; (**B**) p53; and (**C**) caspase-3 mRNA expression levels in a mouse model of ALI induced by CCL_4_. Data are expressed as mean ± SD, *n* = 8. Different letters indicate statistically significant differences between groups at (*p* < 0.05), one-way ANOVA.

**Figure 8 molecules-27-08650-f008:**
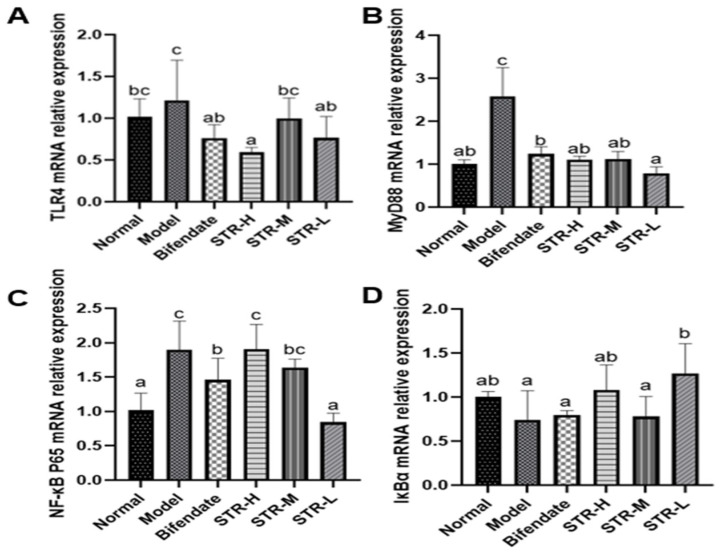
Effects of STR on the mRNA expression of TLR4, MyD88, NF-κB, and IκBα. (**A**) TLR4; (**B**) MyD88; (**C**) NF-κB P65 and (**D**) IκBα mRNA expression levels in a mouse model of ALI induced by CCL_4_. Data are expressed as mean ± SD, *n* = 8. Different letters indicate statistically significant differences between groups at (*p* < 0.05), one-way ANOVA.

**Figure 9 molecules-27-08650-f009:**
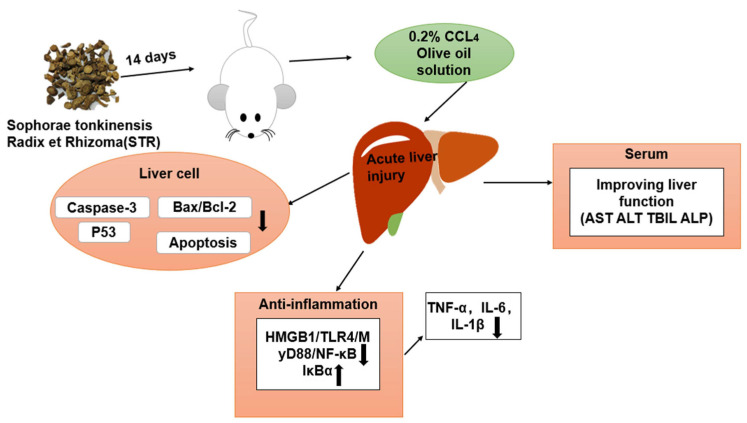
Mechanistic diagram of the protective effects of STR in the murine model of ALI induced by CCL_4_.

**Table 1 molecules-27-08650-t001:** Primer sequences of each gene used in this study.

Gene	Forward (5′-3′)	Reverse (5′-3′)
GAPDH	CAA GAA GGT GGT GAA GCA GGC	CCA GGA AAT GAG CTT GAC AAA G
Bax	TGA GCG AGT GTC TCC GGC GAA T	GCA CTT TAG TGC ACA GGG CCT TG
P53	CGC CGA CCT ATC CTT ACC ATC ATC	GGC AGT TCA GGG CAA AGG AC
Caspase3	TGGGACTGATGAGGAGA	ACTGGATGAACCACGAC
MyD88	TCATGTTCTCCATACCCTTGGT	AAACTGCGAGTGGGGTCAG
NF-κB P65	GCCTCTGGCGAATGGCTTTA	TGCTTCGGCTGTTCGATGAT
IκBα	ATGCCAGAACGAGATAGTGAGC	AGGTGGCGCAGAAGTAGGT
TLR4	TCTGGGGAGGCACATCTTCT	AGGTCCAAGTTGCCGTTTCT

## Data Availability

The data presented in this study are available on request from the corresponding author.

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
