# Peer review of "Protective Effects of Sophorae tonkinensis Gagnep. (Fabaceae) Radix et Rhizoma Water Extract on Carbon Tetrachloride-Induced Acute Liver Injury"

_molecules, 2022, doi:10.3390/molecules27248650_

Round 1

Reviewer 1 Report (Previous Reviewer 2)

The manuscript can be accepted for publication in this form.

Author Response

Thank you for your favorable comment.

Reviewer 2 Report (Previous Reviewer 1)

Author Response

Comment 1: Line 28 the accurate active ingredients of the STR water extract will be explored in near future.

Answer: We have revised this part according to the Reviewer’s suggestions.

Comment 2: Line 36 and 37 “and participates in transformation and toxicant transformation, and plays an important role in maintaining the basic metabolism of the body and other functions” English language needs to be revised by a native speaker.

Answer: This sentence has been revised into “which participates in toxicant transformation and plays an essential role in maintaining the basic metabolism of the body and other functions”.

Comment 3: Line 68 Sophora tonkinensis Gagnep, a legume plant that is mainly distributed in Guangxi, Name should be italic

Answer: We are very sorry for ignoring the formatting of this word. We have revised it in Line 68.

Comment 4: Line 86 and 87 This study not only illustrated the efficacy of STR water extract through scientific theory but also provided the necessary data for further development of STR in the toxic liver injury. Please explain this sentence and how it is supported

Answer: We appreciate for the Reviewer’s good comments. After careful verification, we found that this sentence was redundant and should be deleted. Now it is deleted.

Comment 5: Line 119 and 120 Dried STR (100 g) (Figure 1) was washed with deionized water and STR was then extracted three times using 10×,8×, and 8× deionized water(weight/volume) for 1 h each time. the authors apparently don’t have enough knowledge of chemistry. The extraction method is deeply flawed and cannot extract the present ingredients.

Answer: As the reviewer has mentioned, our extraction method was water method. This study is a preliminary study of the protective effect of Sophorae tonkinensis Radix et Rhizoma water extracts on CCL4 liver-injured mice, so we used deionized water for extraction. Moreover, the STR is a herbal medicine and usually boiled with water to treat liver-related diseases. Meanwhile, STR contains alkaloids, flavonoids, and other active ingredients. Many materials could be extracted by water extraction method at the boiling temperature. Therefore, although the water extraction method is simple, many chemical compositions can be extracted.

Comment 6: Line 135 and 146 o evaluate the material basis of the pharmacological effects of STR, the STR aqueous extract was diluted 100 times with absolute ethanol and centrifuged at 12000 g at 4 ℃ for 10 min to obtain the supernatant. The supernatant was the Same as the previous comment

Answer: As we have mentioned in the answer to the Comment 5, many chemical compositions can be extracted from STR by the water extraction method. The STR aqueous extract was used to determine the chemical ingredients by UPLC-MS/MS and top 20 chemicals were predicted. The whole process meets the study expectation.

Comment 7: Line 246 table 2. | Top 20 components of Sophorae tonkinensis Radix et Rhizoma (STR) water extract by ultra-high-performance liquid chromatography-tandem mass spectrometry analysis. Data shown here are not enough to identify the claimed compounds. If the authors have any knowledge of chemistry, they should have known that the mentioned measurements are not needed to confirm the compounds.

Answer: We are very grateful to the reviewer for the professional chemistry knowledge and comments. Indeed, we are not good at pharmaceutical chemistry and performed the UPLC-MS/MS with the help of a technician. The purpose of this test is to show the fingerprint of Sophorae Tonkinensis Radix et Rhizoma (STR) to the readers as a qualitative reference index. Our description was not accurate in the manuscript, and these 20 top chemical compositions in the water extract of STR were predicted by UPLC-MS/MS. Therefore, we have revised the title of Table 2 into “Predicted top 20 components of Sophorae tonkinensis Radix et Rhizoma (STR) water extract by ultra-high-performance liquid chromatography-tandem mass spectrometry analysis.”.

Round 2

Reviewer 2 Report (Previous Reviewer 1)

the chemistry part is not suitable for publishing. please delete it at all. you cannot mislead readers and students by publishing predictions from an instruments without being revised and confirmed by a chemist.

Author Response

Comment: the chemistry part is not suitable for publishing. please delete it at all. you cannot mislead readers and students by publishing predictions from an instruments without being revised and confirmed by a chemist.

Answer: We agreed with the reviewer’s good suggestions and have deleted the Table 2. Meanwhile, the Part 3.1 was modified accordingly in Lines 235-238.

“3.1. Quality control analysis of STR by UPLC-MS/MS

As shown in Figure 2, UPLC-MS/MS was used to predict the chemical composition of the STR water extract, and the spectra fingerprint of the STR water extract was established as a qualitative reference index.”

This manuscript is a resubmission of an earlier submission. The following is a list of the peer review reports and author responses from that submission.

Round 1

Reviewer 1 Report

Reviewer 1 comments:

The manuscript molecules-2034890 reported the UPLC—MS  of the Sophorae tonkinensis Radix et Rhizoma (STR) and an in-vivo study measuring  serum aminotransferase, (HMGB1)as well as other inflammatory factors. The (TLR4), MyD88, and nuclear factor kappa B (NF-κB) were assessed. The study included many biomarkers to confirm the improvement of liver function, yet the topic is not new and was discussed in previous paper in 2013. The chemistry part is chaotic and contains deep flaws as in the comments below.

He CM, Cheng ZH, Chen DF. Qualitative and quantitative analysis of flavonoids in Sophora tonkinensis by LC/MS and HPLC. Chin J Nat Med. 2013 Nov;11(6):690-8. doi: 10.1016/S1875-5364(13)60081-3. PMID: 24345512.

Comment 1:

The chemistry part is preliminary and the compounds’ identity cannot be confirmed by the measured data. In fact, no need to mention the area% or the retention time at all to identify the compounds. Important data as the retention index and the MSMS are missing.

The authors mention that they isolated and identified the compounds, if they have any chemistry knowledge, they should have known that the UPLC MSMS doesn’t isolate or purify the compounds.

Comment 2

Many artifacts are reported as compounds in the results such as Diisobutylphthalate and 2,2,6,6-Tetramethyl-1-piperidinol (TEMPO)

Comment 3:

I would like to know how the STR extract was prepared. It is only mentioned to be water extract, which is weird. What are the conditions of extraction?

Comment 4:

The chemistry part contradicts totally with the following paper conducted on the same plant in the same country.

He CM, Cheng ZH, Chen DF. Qualitative and quantitative analysis of flavonoids in Sophora tonkinensis by LC/MS and HPLC. Chin J Nat Med. 2013 Nov;11(6):690-8. doi: 10.1016/S1875-5364(13)60081-3. PMID: 24345512.

Comment 5:

The hepatoprotective effect has been already reported before in this articles.

Cai L, Zou S, Liang D, Luan L. Structural characterization, antioxidant and hepatoprotective activities of polysaccharides from Sophorae tonkinensis Radix. Carbohydr Polym. 2018 Mar 15;184:354-365. doi: 10.1016/j.carbpol.2017.12.083. Epub 2018 Jan 2. PMID: 29352929.

Reviewer 2 Report

Dear Dr. Authors

  Thank you very much for choosing this idea and for this good work. Please find the comments.

Title: The scientific name should write in italic form with the author and the plant family.

 Protective Effects of Sophorae tonkinensis Gagnep. (Fabaceae) Radix et Rhizoma Extract on Carbon Tetrachloride-Induced Acute Liver Injury

 Abstract: Should include background, objective, materials and methods, results, and conclusion.

 Introduction:

·       The aim needs to be more precise.

 Results

·       The explanations of the abbreviations in Table 2 should be added.

·       The arrangement of Tables and Figs needs revision.

·       In the Figs. 3, 5, 6, and 7, data are expressed as mean ± SD, while it should be expressed as mean ± SE; to show the error bar. Also, letters should be put above the bars to indicate the differences among the treatments.

·       In Fig. 5a and b, the SD of STR-H is very high. Why??

 Discussion

·       discussion needs more improvements.

 Conclusion

·       The conclusion to be more clear.

 References

·       Scientific names should be in italic form.